# Magnetic reconnection driven by electron dynamics

Y. Kuramitsu [1,2,3], T. Moritaka[3,10], Y. Sakawa[2], T. Morita[2,8], T. Sano [2], M. Koenig[1,4], C.D. Gregory[5], N. Woolsey [6], K. Tomita[7], H. Takabe[8], Y.L. Liu[3], S.H. Chen[3], S. Matsukiyo[7] & M. Hoshino[9]

Magnetic reconnections play essential roles in space, astrophysical, and laboratory plasmas, where the anti-parallel magnetic field components re-connect and the magnetic energy is converted to the plasma energy as Alfvénic out flows. Although the electron dynamics is considered to be essential, it is highly challenging to observe electron scale reconnections. Here we show the experimental results on an electron scale reconnection driven by the electron dynamics in laser-produced plasmas. We apply a weak-external magnetic field in the direction perpendicular to the plasma propagation, where the magnetic field is directly coupled with only the electrons but not for the ions. Since the kinetic pressure of plasma is much larger than the magnetic pressure, the magnetic field is distorted and locally anti-parallel. We observe plasma collimations, cusp and plasmoid like features with optical diagnostics. The plasmoid propagates at the electron Alfvén velocity, indicating a reconnection driven by the electron dynamics.

[1] Graduate School of Engineering, Osaka University, 2-1 Yamadaoka, Suita, Osaka 565-0871, Japan. [2] Institute of Laser Engineering, Osaka University, 2-6 Yamadaoka, Suita, Osaka 565-0871, Japan. [3] Department of Physics, National Central University, No. 300, Jhongda Rd., Jhongli 32001 Taoyuan, Taiwan. [4] LULI - CNRS, Ecole Polytechnique, CEA: Université Paris-Saclay, UPMC Univ Paris 06:Sorbonne Universités, 91128 Palaiseau cedex, France. [5] Rutherford Appleton Laboratory, Chilton, Didcot OX11 0QX, UK. [6] York Plasma Institute, Department of Physics, University of York, York YO10 5DD, UK. [7] Faculty of Engineering Sciences, Kyushu University, 6-1 Kasugakoen, Kasuga, Fukuoka 816-8580, Japan. [8] Helmholtz-Zentrum Dresden-Rossendorf, Bautzner Landstr. 400, 01328 Dresden, Germany. [9] Department of Earth and Planetary Science, University of Tokyo, 7-3-1 Hongo, Bunkyo, Tokyo 113-0033, Japan. [10] Present address: Department of Helical Plasma Research, National Institute for Fusion Science, Toki 509-5292, Japan. Correspondence and requests for materials should be addressed to Y.K. (email: kuramitsu@eei.eng.osaka-u.ac.jp)

Magnetic reconnection is a fundamental factor in space and astrophysical plasmas, where anti-parallel magnetic field components reconnect and release the magnetic energy as the plasma kinetic energy[1–3]. It governs various phenomena such as magnetospheric substorms, stellar and solar flares, and their winds and coronal heatings. The magnetic reconnection can also play essential role in particle acceleration in the presence of collisionless shocks[4,5]. It has been widely believed that the electron dynamics is essential in the triggering processes of the magnetic reconnections[6–8], however, it is highly challenging to observe the electron features due to their small spatial and temporal scales both in space and laboratory plasmas[9,10]. The first electron scale measurements of magnetic reconnection in space plasma were reported very recently[10]. However, as the space observations are provided with a (few) spacecrafts in situ, there is no global observation or imaging of magnetic reconnections. With the in situ observations the structure of the X-line and jets are provided with numerical simulations or schematic figures. On the other hand, imaging of solar flares provide us global structures of magnetic reconnections such as cusp and plasmoid[11], although it is impossible to resolve the electron scale dynamics. Even though it is difficult to simultaneously observe electron and ion scale phenomena due to the large scale difference between these two species, we can properly model some aspects of the physical processes, where the electron kinetics governs the macroscopic structures in laboratory.

The magnetic reconnections have also been investigated with magnetic confinement device[1]. While in space plasmas a single or a few observation points provide the magnetic field data, the magnetic reconnection experiments in the laboratory provide multiple possible observations, which allow the imaging of the reconnection regions. Recently, fast reconnections have been observed in laser-produced plasmas[12–18]. In these experiments, multiple beams are focused on a solid target with small separations, and then the baroclinic magnetic fields are generated surrounding the laser-produced plasmas, resulting in magnetic reconnection with the anti-parallel field geometry. These reconnections are strongly driven by the expanding plasmas that collide each other[19]. Our experimental approach is completely different from the previous studies with high-power lasers, where the laser–matter interactions and the self-generated magnetic fields play an essential role. We focus on electron dynamics by using a weak external field produced by a permanent magnet. Under the influence of this field, that is perpendicular to the plasma flow propagation axis, it can be collimated due to the distortion of the

magnetic field[20]. We adjust the field strength so that the electrons are magnetized but not the ions, while we keep the system size much larger than the ion inertial length, $L_s \gg c/\omega_{pi}$, where $c$ is the speed of light and $\omega_{pi}$ is the ion plasma frequency. This system is directly applicable to the magnetic reconnection driven by the electron dynamics.

Here we report the experimental results on an electron scale reconnection governed by the electron dynamics in laser-produced plasmas. We separate the electron scale from the ion scale by a weak external magnetic field. The magnetic field is strong enough to magnetize the electrons but not the ions; the magnetic field is not directly coupled with ions. We have observed thin plasma structures and plasmoid like features with optical diagnostics, indicating a reconnection at microscopic scale driven by the electron dynamics. The plasmoid is accelerated due to the reconnection up to the electron Alfvén velocity, indicating efficient acceleration of electrons.

## Results

**Experiment**. Figure 1(a, b) schematically show the laser and target configurations without and with an external magnetic field, respectively (see more details in Methods). Figure 1(c, e) show interferograms of the plasmas in the absence of the external magnetic field at 8 ns and 15 ns after the main laser irradiation, respectively. The rear-side plasma unloaded in vacuum longitudinally rather than transversely due to the beam offset; the lateral size of the plasmas at 15 ns was similar to the one at 8 ns. Figure 1(d) and the upper half of 1(f) show interferograms in the presence of the magnetic field at 10 ns and 15 ns, respectively. The lower half of Fig. 1(f) shows the electron density map obtained from the interferogram. The electron collimation is clearly seen on the propagation axis; the lateral size being much smaller than the one in the absence of the magnetic field.

From the snapshot of the electron density map we estimate the plasma velocity; the fringe shifts are recognized at 8 mm from the target in 15 ns in Fig. 1(f), thus, the fast plasma velocity is about $500 \, \text{km s}^{-1}$. This is consistent with the SOP result (not shown). Using this as the ion bulk velocity, the ratio between the kinetic pressure to the magnetic pressure is estimated as $\beta_K \equiv n_i m_i v_i^2 \mu_0 / B^2 \sim 10^5$, where $n_i$ is the ion density, $m_i$ is the ion mass, $v_i$ is the ion velocity, $\mu_0$ is the magnetic permeability in vacuum, and we assume average mass and charge of protons and fully ionized carbon ions. Even though the kinetic energy is much larger than the magnetic energy, such weak magnetic field affects

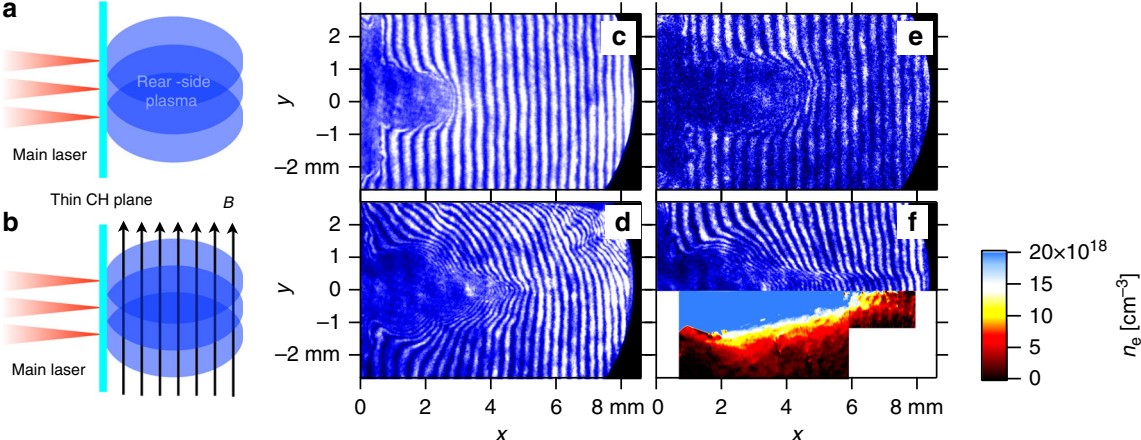

**Fig. 1** Electron density distributions with and without an external magnetic field. Schematic of the plastic (CH) planar target: **a** without an external magnetic field and **b** with an external magnetic field. The laser beam offset is 100–200 µm. Interferograms without the external magnetic field **c** at 8 ns and **e** at 15 ns. Interferograms with the external magnetic field **d** at 10 ns and **f** at 15 ns

the plasma propagation. With $v_i = 500$ km s$^{-1}$ and $B = 0.3$ T the proton gyroradius is $r_{gi} \equiv v_i m_p/(eB) \sim 17$ mm (much larger than our system), where the $e$ is the element charge and $m_p$ is the proton mass, and that of the carbon ion with its charge of $Z = 6$ and mass of $m_c = 12 m_p$ is simply twice of the proton's. Besides, the electron gyroradius with the same velocity is $r_{ge} \sim 9.5 \mu m$, so electrons are well magnetized in our system and cannot freely propagate across the magnetic field. Note that the ion velocity is estimated with the interferogram, however, the ions are not magnetize in our system, the ion velocity can be larger than our estimation. These estimations provides minimums.

**Plasma collimation**. A simple explanation for the plasma collimation is schematically shown in Fig. 2 (and also numerically confirmed in the supplementary information as shown in Supplementary Figure 1). In the presence of a weak magnetic field (**B**) in vacuum a plasma propagates in the direction perpendicular to the magnetic field. When the kinetic pressure is much larger than the magnetic pressure, the magnetic field will be stretched by the plasma. Stretching the field induces a new magnetic field as in Fig. 2(a). In our system only electrons are trapped by the magnetic field, i.e., the space charge will be generated. Consequently an electrostatic field (**E**) is excited across the distorted magnetic field [Fig. 2(b)]. In the presence of the electric and magnetic fields, the electrons move in the direction perpendicular to the both fields due to the **E** × **B** drift. In a macroscopic system the **E** × **B** drift does not carry a net current, however, in our microscopic system, only the electrons are magnetized and produce this. As a result, a finite current (**J**) carried by the electrons is generated in the system [Fig. 2(b)], which has to be self-consistent with the magnetic field distortion. While the induced field prevents the transverse plasma expansion, the stretched field is more parallel to the propagation axis, i.e., the plasma can freely propagate only along the plasma axis [Fig. 2(b)]. Therefore, there are positive feedbacks to make plasma further collimated or thiner.

If the field is elongated, the local field above and below the plasma axis is anti-parallel. So, if one can stretch the field long enough, such as the Earth's magnetotail or floating flux tube on the Sun, magnetic reconnection is possible. In order to further enhance the reconnection possibility, we add an ambient medium (nitrogen gas) in the target chamber. This gas is ionized prior to the jet arrival by the X-ray radiation coming from the laser–matter interaction. The ambient plasma is pushed by the jet and a bow shock is formed in the ambient plasma. The shocked ambient plasma provides an external pressure on the jet [Fig. 2(c)].

Figure 3(a) shows a schematic image of the plasma topology associated with a magnetic reconnection. The plasma flow elongates the magnetic fields and the anti-parallel magnetic field lines can reconnect due to the external pressure of the shocked ambient plasma. The reconnection transforms the magnetic energy as plasma kinetic energy. The leading edge of the plasma is detached from the jet and is released as a plasmoid. The cusp is an acute structure often seen in magnetic reconnection[11]. Figure 3(b) shows a two spatial dimensions (2D) snapshot image of self-emission at 35 ns after the main laser beams firing. The laser is coming from the left and the nominal focal spot is located at $(x, y) = (0, 0)$. Since the fast rear-side plasma unloades in the ambient plasma, a bow shock is generated and clearly observed. The faint structure at $(1, 0)$ is a shock wave going from the other side the target ($x < 0$) around to the rear side ($x > 0$). Behind the shock, thin structures are evidently seen on the plasma propagation axis ($0 \lesssim x \lesssim 6$ mm and $y = 0$). These structures are separated into two at $(x, y) \sim (3.6, 0)$, where cusp like structures

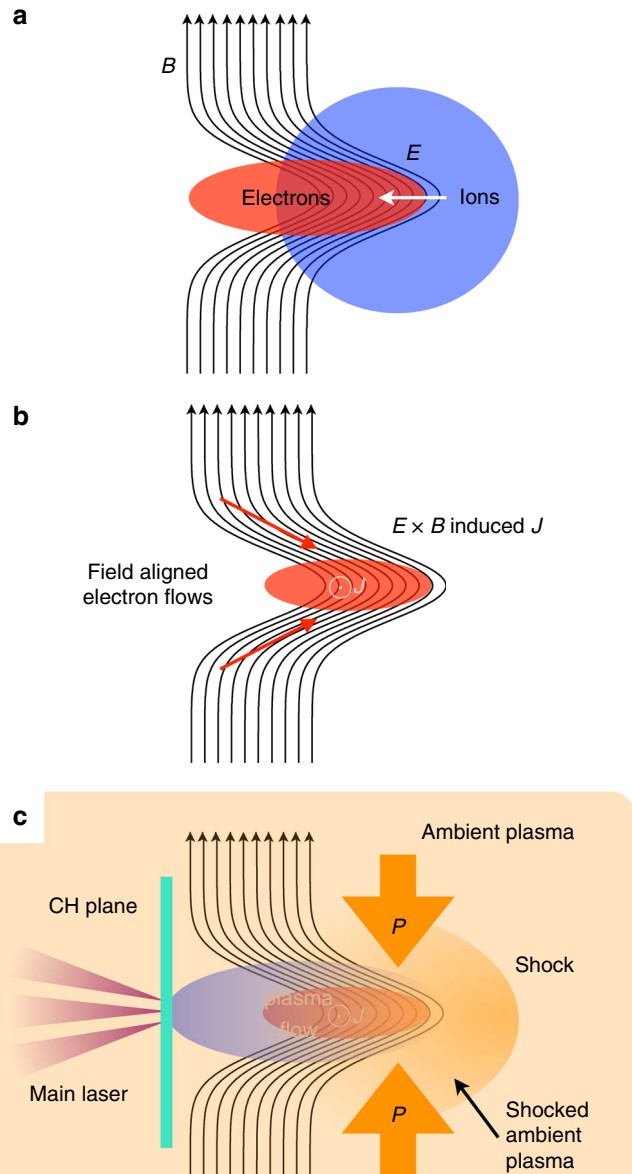

**Fig. 2** Model (**a**) A directional plasma flow propagates in the presence of a weak-perpendicular magnetic field, where the kinetic energy is much larger than the magnetic energy and only electrons are directly coupled with the magnetic field, resulting in distortion of the magnetic filed **B** and the charge separation producing an electric field **E**. **b** **E** × **B** generated a finite electron current **J** since the ions are not magnetized at this scale. In the distorted field lines, electrons tend to propagate along the magnetic field. This enhances the field distortion and current. **c** By adding an ambient medium, an external pressure is provided to the plasma flow

form. Some plasma is separated from the main plasma flow, i.e., a plasma island or plasmoid is formed.

Figure 4(a) show a schematic image of the time evolution of reconnection. Elongated magnetic field lines release their tension as outflows of a magnetic reconnection. The outflow velocity or the plasmoid and also the rear-side plasma velocity with respect to the reconnection point is considered to be of the order of the Alfvén speed $c_A$. Therefore, the separation velocity in Fig. 4(a), $\Delta v$ has to be of the order of $c_A$. Figure 4(b) shows the time evolution of the plasma on the propagation axis, obtained with the SOP diagnostic. The fastest structure is the bow shock. Behind the

shock, one can see that the plasmoid is detached from the plasma flow. The arrows 1 and 2 correspond to $115\,km\,s^{-1}$ and $16\,km\,s^{-1}$, respectively, resulting in $\Delta v = 50$, which is the relative velocity and independent of choice of the reference frame.

## Discussion

In Fig. 1(f) the collimated electron flow in the presence of the external magnetic field extends further than that without the external field in Fig. 1(e). There are two possible explanations; (1) the plasma is faster in the presence of the external perpendicular magnetic field that that in the absence of the field, and (2) since the electron is collimated in the presence of the external field, the central electron density is higher than that without the external field. Interferometry has a certain detective range of electron density [e.g.,ref.[21]]. In the latter case, it is not necessary to assume that the plasma is actually faster in the presence of the external magnetic field. This can be confirmed with our simulations in Supplementary Figures 2(a) and 2(b), where the electrons are collimated in the presence of the external magnetic field while the low density electrons propagate further in the absence of the field. Furthermore, in the presence of an ambient plasma, where a shock exists, the shock velocity is slightly faster in the absence of the external magnetic field than that in the presence of magnetic field as in Fig. 4(b) and Supplementary Figure 3. This clearly shows that the plasma velocity in the presence of the external magnetic field is similar to or slower than that in the absence of the magnetic field. We consider the latter case in this paper. In this scenario, the electron plasma in the presence of the magnetic field does not extend further than the case in the absence of the magnetic field. This is completely evident in Supplementary Figure 2, where the simulations show that the magnetized electrons are significantly retarded compared with the unmagnetized case. Supplementary Figure 3 also demonstrates that the magnetized plasma does not move faster than the unmagnetized case.

Cusps and plasmoids are key features of the reconnection. Plasmamoids play a fundametal role in the fast reconnections[22–24]. If this plasmoid propagates at Alfvén velocity, one can conclude that the plasmoid results from the reconnection. In our system only electrons are directly coupled with the magnetic field, and thus, the outflow velocity is relevant to the electron dynamics. The electron Alfvén speed is defined with the electron mass as $c_{Ae} = B/\sqrt{\mu_0 n_e m_e}$. In order to estimate this, we use the initial magnetic field $B = 0.3\,T$ and the electron density from Fig. 1(f). We plotted the electron density in Fig. 1(f) up to $2 \times 10^{19}\,cm^{-3}$ but the

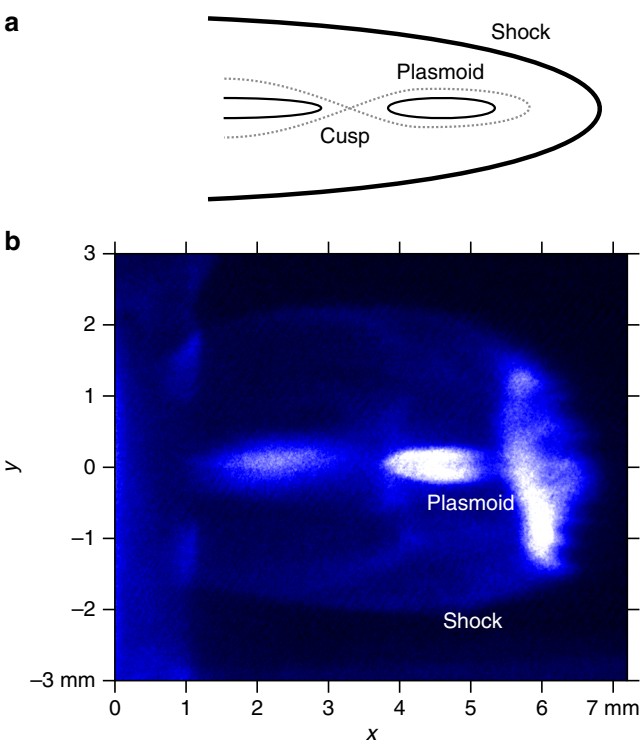

**a**

**b**

**Fig. 3** Two dimensional snapshot of cusp and plasmoid. **a** Schematic view of a magnetic reconnection. **b** Image of self-emission obtained with gated charge coupled devise (CCD) camera at 35 ns after the main laser shot. The target environment is gas-filled (5 Torr Nitrogen)

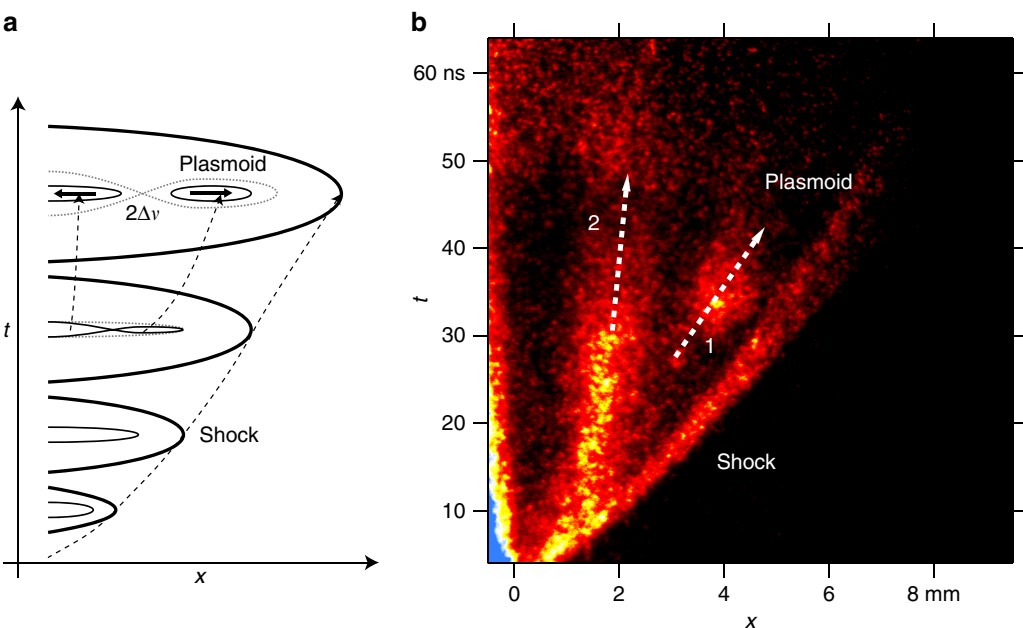

**a**

**b**

**Fig. 4** Time evolution of shock, plasma, and plasmoid. **a** Schematic Image of the time evolution of the magnetic reconnection. **b** Streaked self-emission optical pyrometer (SOP) image. The white arrows 1 and 2 indicate the velocities of the plasma structures

collimated electron density is about $2-5 \times 10^{19}$ cm$^{-3}$. We cannot obtain the density from the interferogram of the same shot as Fig. 4 (b), since the fringes are discontinues due to the shock wave. Although these are different shots and different conditions, the density in Fig. 1(f) can be considered as a reference of the density of rear-side plasma in Fig. 4(b). Using $n_e = 2-5 \times 10^{19}$ cm$^{-3}$ at the plasma propagation axis in Fig. 1(f), the electron Alfvén speed $c_{Ae} = 40-63$ km s$^{-1}$. This is very close the outflow velocity in Fig. 4(b).

In this paper we have discussed a magnetic reconnection driven by electron dynamics in a laser-produced plasma. In the presence of a weak-perpendicular magnetic field, where only electrons are magnetized, plasma collimation is observed. Since the ions in our system are not magnetized, the structure formation solely observed in the presence of the field is an evidence that the electrons dynamics controlling the macroscopic structure. Our experimental results with the ambient plasma show the essential aspects of reconnections, i.e., the cusp and the Alfvénic outflows. These results bring tremendous benefit to various fields of science and engineering, such as magnetic sails[25–27], micro magnetospheres[28–31], plasma jets[21, 32–34], and the reconnections in electron scales[6–8]. For instance, lunar surface magnetic fields have typically smaller size than ion gyroradius of the solar wind; it is considered that the electron dynamics plays essential roles on their interactions with the solar wind[29–31].

## Methods

**Experimental conditions and diagnostics**. The experiment was performed with Gekko XII HIPER laser system at the Institute of Laser Engineering, Osaka university. The main laser beams (130 J energy each at 351 nm wavelength, a Gaussian pulse length of 500 ps, and a 300 μm diameter focal spot) were used to irradiate a plastic plannar (10 μm of CH, 3 mm × 3 mm square size) to produce a fast plasma in the rear side of the target. Plasma is also created on the side of laser irradiation, we mainly focus on the rear-side plasmas. Figures 4(a, b) show a schematic drawing of the target without and with an external magnetic field, respectively. A permanent magnet ($B \sim 0.7$ T at the surface) was located ~5 mm below the target center, giving a magnetic field $B \sim 0.3$ T perpendicular to the target normal. Since the diameter of the magnet was 30 mm, the field where the plasma propagates was effectively uniform. The laser beams were separated by 100–200 μm in order to make the plasma directional[21]. The spatial and temporal evolutions of the rear-side plasmas were observed by transverse optical diagnostics with 2D time resolved detectors (self-emission and interferometry) and streaked optical pyrometer (SOP). The interferometer was a modified Nomarski type[35], and the self-emission was taken at the wavelength of 450 nm.

**Code availability**. The custom particle-in-cell code used to generate the simulation results in the current study is available from the corresponding author on reasonable request.

## Data availability

The data sets generated during and/or analysed during the current study are available from the corresponding author on reasonable request.

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

## Acknowledgements

The authors would like to thank S. Imada, K. Kusano, and the technical support by the staff at the ILE. This work was supported by the Ministry of Science and Technology, Taiwan under Grant No. MOST-103 -2112 -M-008-001-MY2, MOST 104-2911-I-008-504, and

MOST 105-2112-M-008 -003 -MY3; JSPS KAKENHI Grant Number 24740369 and 21340172; a grant for the Core-to-Core Program from JSPS; and the Asian Core Program for High energy density science Using intense Laser photons commissioned by JSPS.

## Author contributions

Y.K., Y.S., T.Morita., C.D.G, M.K., and N.W. carried out the Gekko experiment. Y.K. analyzed the data. T.Moritaka. performed the numerical simulations. Scientific discussions were provided by T.S., K.T., Y.L.L., S.H.C., S.M., H.T., and M.H.. The paper was written by Y.K.

## Additional information

**Competing interests:** The authors declare no competing interests.

