## [Peer Review File · Nature Communications]

Reviewers' comments:

Reviewer #1 (Remarks to the Author):

I confirm my previous positive report written for Nature Physics and acknowledge that the authors modified the paper according to my remarks.

The paper is in my opinion suitable for publication on Nature Communications.

I noted a misprint on page 2 where plasmamoids is written instead of plasmoids.

Reviewer #2 (Remarks to the Author):

I read all comments and responses and I'm not sure why I was even sent this. Obviously it is ready for publication. I hardly see even the need for another round of refereeing. My recommendation is to publish as is.

Reviewer #3 (Remarks to the Author):

Second Referee report on (revised manuscript 'Magnetic reconnection driven by electron dynamics' by Y. Kuramitsu et al.

I have read the above reference revised manuscript and the rebuttal provided by the authors to the previous reports by the 3 referees. As is my habit, I will confine myself to comments on whether the authors have adequately addressed the concerns I raised in my initial report.

A. Summary of the key results. As I previously reported, this paper reports a new observation which the authors claim represents magnetic reconnection occurring within a laser-produced plasma in a laboratory setting. The experimental set-up involves the production of a collimated plasma beam which propagates into a weak magnetic field. The strength of the field is such that the electrons produced in the plasma are likely to remain magnetized, while the ions are not. A key aspect of the initial paper was the interpretation of interferograms, from which the authors claimed the ions appear to propagate ahead of the electrons, which deform the field. Some time after the launch of the plasma, a detached blob is detected which appears to move with the anticipated Alfvén speed of the plasma, and therefore is consistent with expectations of reconnection outflow jets and plasmoid formation.

B. Originality and significance: As I previously judged, the observations reported are original, and the overall interpretation may be correct that the morphology of the plasmas observed is related to the process of magnetic reconnection. That being the case, the observations would be of interest to the broad community of plasma physicists.

C. Data & methodology: In my previous report I raised a number of concerns which I felt would preclude the paper from being acceptable for publication in any journal. To their credit, the authors have, I feel, tried to address my concerns in the revised version. However, I also feel that this is with mixed success, and so I have remaining (and perhaps even renewed) doubts about the validity of the interpretation.

A significant concern from my previous report was that the interpretation appeared to be inconsistent

with the observations presented in Figure 1, where the plasma appears to move FASTER in the presence of the magnetic field, not slower. In their rebuttal, the authors claim that this is 'simply' due to the fact that interferometry has a certain detective range of electron density and that since the electron is collimated in the presence of the external field, the central electron density is higher than that without the external field. Thus they claim that the plasma appears to extend further in the presence of the external field, but this is not actually the case. I have 2 big problems with this. Firstly and most fundamentally, if we cannot actually infer real boundaries, or plasma extents, from the images, how can we make any valid inference on what the images mean at all? To me this argument by the authors seems to pull the rug out from underneath the whole paper, so it would be a further reason not to recommend the paper for publication! However, I cannot actually see how this could be correct, or at least I cannot see how the claim that this is supported by the simulation in the supplementary material can be correct. If I look at the top half of Figure 5, panels (a) and (b), I cannot see how I can select any value for a limit to the 'detective range of electron density' which would reproduce the pattern shown in the observations. For example, I might most obviously choose a value for detectability corresponding to $\log(n_e/n_o) > -4$ (the clear boundary between black and blue), but then the contour for the detection of plasma will NOT show the pattern the authors claim is due to this effect. The electron plasma in the magnetic field does not extend further than the non-magnetic field case, nor can I see how any other choice would give you this result. Thus I feel that the authors have just muddied the waters and the issue raised in my first report remains a concern and an impediment to publication.

Given the above answer now introduces uncertainty in the extent of any plasma flow, I feel a second major point from the previous review remains unaddressed. I specifically stated 'case from the observations since it is not made clear which distortions in Figure 1 relate to the (unmodified?) ion flow and which to the (retarded?) electron flow. In other words, on what basis are the extents of the 2 populations drawn in Figure 2b determined from the data?' The answer in the response seems to be that these boundaries are actually inferred from the simulation, not the observation, which again questions the value and relevance of the latter.

On further major point from the previous review, that the physical reasoning applied to explain the remainder of Figure 2 appeared flawed, the authors have improved the figure and the descriptions. They now explicitly recognise that the current produced is simply that which self consistently supports the field distortion and have deleted the erroneous $J \times B$ panel. The physics of this is correctly described now I think.

Finally, the discussion of the plasmoid velocity was improved but I would still prefer that the authors should at least tell us IN THE PAPER how these underlying parameter values were determined from the experiment?

D. Appropriate use of statistics and treatment of uncertainties: See statement above regarding lack of detail/explanation of how experimental parameters were determined.

E. Conclusions: robustness, validity, reliability: As before, I have a feeling that there are still significant issues with the detailed interpretations which thus cast doubt on the validity of this conclusion in their current form. The new statements on the relevance of the interferometry images make me less sure of the relevance of the observations, assuming they are even correct in terms of their application here!

F. Suggested improvements: experiments, data for possible revision: As before, I think the paper needs work to provide much more robust evidence for its conclusions before it can be recommended for publication.

G. References: appropriate credit to previous work? This is still OK.

H. Clarity and context: lucidity of abstract/summary, appropriateness of abstract, introduction and conclusions: This is still OK, the material presented remains succinctly written.

We highly appreciate the referee's comments and suggestions that further improve our manuscript. We have revised our manuscript based on the reviewer's comments. The revision is written in red in the manuscript.

Reviewer #3 (Remarks to the Author):

Second Referee report on (revised manuscript 'Magnetic reconnection driven by electron dynamics' by Y. Kuramitsu et al.

I have read the above reference revised manuscript and the rebuttal provided by the authors to the previous reports by the 3 referees. As is my habit, I will confine myself to comments on whether the authors have adequately addressed the concerns I raised in my initial report.

A. Summary of the key results. As I previously reported, this paper reports a new observation which the authors claim represents magnetic reconnection occurring within laser-produced plasma in a laboratory setting. The experimental set-up involves the production of a collimated plasma beam which propagates into a weak magnetic field. The strength of the field is such that the electrons produced in the plasma are likely to remain magnetized, while the ions are not. A key aspect of the initial paper was the interpretation of interferograms, from which the authors claimed the ions appear to propagate ahead of the electrons, which deform the field. Some time after the launch of the plasma, a detached blob is detected which appears to move with the anticipated Alfvén speed of the plasma, and therefore is consistent with expectations of reconnection outflow jets and plasmoid formation.

B. Originality and significance: As I previously judged, the observations reported are original, and the overall interpretation may be correct that the morphology of the plasmas observed is related to the process of magnetic reconnection. That being the case, the observations would be of interest to the broad community of plasma physicists.

C. Data & methodology: In my previous report I raised a number of concerns which I felt would preclude the paper from being acceptable for publication in any journal. To their credit, the authors have, I feel, tried to address my concerns in the revised version. However, I also feel that this is with mixed success, and so I have remaining (and perhaps even renewed) doubts about the validity of the interpretation.

A significant concern from my previous report was that the interpretation appeared to be inconsistent with the observations presented in Figure 1, where the plasma appears to move FASTER in the presence of the magnetic field, not slower. In their rebuttal, the authors claim that this is 'simply' due to the fact that interferometry has a certain detective range of electron density and that since the electron is collimated in the presence of the external field, the central electron density is higher than that without the external field. Thus they claim that the plasma appears to extend further in the presence of the external field, but this is not actually the case. I have 2 big problems with this. Firstly and most fundamentally, if we cannot actually infer real boundaries, or plasma extents, from the images, how can we make any valid inference on what the images mean at all? To me this argument by the authors seems to pull the rug out from underneath the whole paper, so it would be a further reason not to recommend the paper for publication! However, I cannot actually see how this could be correct, or at least I cannot see how the claim that this is supported by the simulation in the supplementary material can be correct. If I look at the top half of Figure 5, panels (a) and (b), I cannot see how I can select any value for a limit to the 'detective range of electron density' which would reproduce the pattern shown in the observations. For example, I might most obviously choose a value for detectability corresponding to $\log(n_e/n_o) > -4$ (the clear boundary between black and blue), but then the contour for the detection of plasma will NOT show the pattern the authors claim is due to this effect. The electron plasma in the magnetic field does not extend further than the non-

magnetic field case, nor can I see how any other choice would give you this result. Thus I feel that the authors have just muddied the waters and the issue raised in my first report remains a concern and an impediment to publication.

Authors:

We appreciate further concerns raised by the referee. Our explanations were not clear enough.

First of all, there is no clear boundary of plasma (except in the presence of shock waves), however, there is a clear difference in the observed plasma structures with and without the external magnetic field. That is our focus in the manuscript. We are not focusing on the "plasma velocities" that might be faster in the presence of the external magnetic field.

As the referee pointed out, we also thought that the plasma velocity was faster in the presence of the external magnetic field when we first looked at the data during the experiment. At that time we tried to directly connect the "faster" velocity to magnetic reconnection, however, there was no further evidence to prove the reconnection. That's why we conducted the follow up experiment with the ambient plasma.

As we reported in the previous round, interferometry is sensitive to a certain range of electron density; if the plasma is collimated, the same density region can be extended in space. (This can be seen in the simulation data too. We will come back to this later.) Unlike simulations, the information we can obtain from experiments is limited; what we can observe from the interferograms is the electron distribution with a certain density range at the observed moment. Thus, the velocity of the contour line of the minimum density observable with interferometry is indeed faster in the presence of the external magnetic field. This is not contradictory to the simulation results and also our model.

We also added the supporting simulation data showing contours at the same density in the presence and absence of the external magnetic field. Figs. 6 (a) and (b) show the magnified views of the upper halves (electron density) of Figs. 5 (a) and (b) with different color and the contour lines at the same density. Since the color scale of Fig. 5 is logarithmic, it is hard to recognize what we described in the previous round. With interferometry we can observe the density higher than the detectable threshold determined by the resolution of imaging system. Suppose that the detectable density corresponds to the contour lines, the plasma with the external magnetic field extends further than that without the magnetic field. In Fig. 1 (e) if we can collimate the plasma, we can extend the detectable region as in Fig. 1 (f). There is certainly a possible explanation why the plasma extends further in the presence of the external magnetic field.

It is still possible to assume that the plasma velocity is faster in the presence of the external magnetic field in Fig. 1. However, we were not able to think of how the plasma can be faster in the presence of the perpendicular external magnetic field. Furthermore, there is no evidence of plasma acceleration across the magnetic field in the numerical simulation as well as in the experimental data. We consider that the plasma with the detectable density extends further in the presence of the external magnetic field than the case without the magnetic field, and that the plasma velocity is not faster in the presence of the magnetic field. We described this in the manuscript.

There is another supporting material on the plasma velocities with and without magnetic field. We mentioned earlier that there is no clear boundary of plasmas, however, there is a clear boundary when a shock wave is excited. In Fig. 7 we show a similar image to Fig. 4 (b) except there is no external magnetic field. The shock velocity is slightly faster in Fig. 7 than that in Fig. 4 (b). This clearly shows that the plasma velocity in the

presence of the external magnetic field is similar to or slower than that in the absence of the magnetic field.

In Fig. 7 there is no signal after 57 ns since the tiger timing is 14 ns earlier than that of Fig. 4 (b) to confirm the laser timing ($t=0$ ns, not shown). One can see lower sensitivity of the camera from ~ 20 to ~ 35 ns in Fig. 7, which corresponds to ~ 34 to ~ 49 ns in Fig. 4 (b). The lower sensitivity is due to the fact that these correspond to the central region of CCD where we normally use the alignment of the optical system. These are technical details and nothing to do with our model and interpretation. We just want to clarify the details and emphasize that the plasmoid like feature is in this less sensitive region; it should be more clear if the streak camera is new. Most importantly there is no plasmoid like feature in the absence of the external magnetic field.

Referee:

Given the above answer now introduces uncertainty in the extent of any plasma flow, I feel a second major point from the previous review remains unaddressed. I specifically stated 'case from the observations since it is not made clear which distortions in Figure 1 relate to the (unmodified?) ion flow and which to the (retarded?) electron flow. In other words, on what basis are the extents of the 2 populations drawn in Figure 2b determined from the data?' The answer in the response seems to be that these boundaries are actually inferred from the simulation, not the observation, which again questions the value and relevance of the latter.

Authors:

What we can measure with interferometry is the electron density at a certain range, and Fig. 1 shows that a clear difference on the electron distributions in the presence and in the absence of the external magnetic field. (We cannot state the unmodified ion flow or the retarded electron flow from the experimental data. That is, in our sense, overstatement. This is experimental data; it is not simulation data where we can find all the details.)

Another important fact in the experiment is that the external magnetic field is too weak to magnetize the ions but strong enough to magnetize the electrons.

Furthermore, the kinetic energy of plasma is much larger than that of the magnetic field. From these facts, we construct a model, Fig. 2, which is able to explain the observational facts.

We emphasized the fact that the ions are not magnetized just before the paragraph introducing the model.

In order to support the model, we conducted the numerical simulations. We cannot measure everything from the experiment; we still need some support from numerical simulations as the MMS Science paper still needs the global information from the simulations [Burch et al. 2016, Ref. 10 in the manuscript].

We explained this in the beginning of the supplement information.

Referee:

On further major point from the previous review, that the physical reasoning applied to explain the remainder of Figure 2 appeared flawed, the authors have improved the figure and the descriptions. They now explicitly recognise that the current produced is simply that which self consistently supports the field distortion and have deleted the erroneous $J \times B$ panel. The physics of this is correctly described now I think.

Finally, the discussion of the plasmoid velocity was improved but I would still prefer that the authors should at least tell us IN THE PAPER how these underlying parameter values were determined from the experiment?

Authors:

We appreciate the referee's suggestion. We added all the information in the manuscript.

Referee:

D. Appropriate use of statistics and treatment of

uncertainties: See statement above regarding lack of detail/explanation of how experimental parameters were determined.

E. Conclusions: robustness, validity, reliability: As before, I have a feeling that there are still significant issues with the detailed interpretations which thus cast doubt on the validity of this conclusion in their current form. The new statements on the relevance of the interferometry images make me less sure of the relevance of the observations, assuming they are even correct in terms of their application here!

F. Suggested improvements: experiments, data for possible revision: As before, I think the paper needs work to provide much more robust evidence for its conclusions before it can be recommended for publication.

G. References: appropriate credit to previous work? This is still OK.

H. Clarity and context: lucidity of abstract/summary, appropriateness of abstract, introduction and conclusions: This is still OK, the material presented remains succinctly written.

Authors:

We have made a major revision including new figures in order to address the referee's comments and suggestions. Finally we would like to thank the referee again; his/her comments and suggestions substantially improved our paper. We believe that we clarify all the concerns and that now the paper is suitable for publication.

REVIEWERS' COMMENTS:

Reviewer #3 (Remarks to the Author):

Third Referee report on (revised manuscript 'Magnetic reconnection driven by electron dynamics' by Y. Kuramitsu et al.

I have read the above referenced re-revised manuscript and the rebuttal provided by the authors to my previous second report. The authors have revised one key aspect of the paper, so much of my previous report (summary, originality and significance, etc) has not changed from before. I will confine myself only to comments on the changes the authors have made since my second report.

A. Summary of the key results. As I previously reported, this paper reports a new observation which the authors claim represents magnetic reconnection occurring within a laser-produced plasma in a laboratory setting. The experimental set-up involves the production of a collimated plasma beam which propagates into a weak magnetic field. The strength of the field is such that the electrons produced in the plasma are likely to remain magnetized, while the ions are not. A key aspect of the initial paper was the interpretation of interferograms, from which the authors argued the ions appear to propagate ahead of the electrons, which deform the field. Some time after the launch of the plasma, a detached blob is detected which appears to move with the anticipated Alfvén speed of the plasma, and therefore is consistent with expectations of reconnection outflow jets and plasmoid formation.

B. Originality and significance: As I previously judged, the observations reported are original, and the overall interpretation may be correct that the morphology of the plasmas observed is related to the process of magnetic reconnection. That being the case, the observations would be of interest to the broad community of plasma physicists.

C. Data & methodology: In my previous reports I raised a concern about the interpretation, which appeared to be inconsistent with the observations presented in Figure 1, where the plasma appears to move FASTER in the presence of the magnetic field, not slower. In their original rebuttal, the authors claim that this is 'simply' due to the fact that interferometry has a certain detective range of electron density and that since the electron is collimated in the presence of the external field, the central electron density is higher than that without the external field. Thus they claim that the plasma only APPEARS to extend further in the presence of the external field, but this is not actually the case. I still feel a little unsure about drawing conclusions which rely on a non-measurement. However, it is clear from the new simulation results included in the latest revision, that it is in fact possible to identify a density contour, which, if it is the limit of measurement, would explain the observations. However, the authors still need to be very careful in their text to make it clear that in this scenario the electron plasma in the magnetic field does NOT extend further than the non-magnetic field case. This is completely evident in the new Figure 6, in which the simulations show that the magnetized electrons are significantly retarded compared to the unmagnetised case. This only appears so as an purported artefact of the measurement set-up. The new plot in Figure 7 is also very useful in demonstrating that the magnetised plasma does not move faster than the unmagnetised case.

D. Appropriate use of statistics and treatment of uncertainties: OK.

E. Conclusions: robustness, validity, reliability: As said above, the authors have demonstrated in their new simulation material that there is a defensible interpretation for the non-measurement of the electrons in the unmagnetised case. Although I find it somewhat unsatisfactory to explain the observations with the claim of an 'inability to measure', it seems to me that the authors have addressed the issue and I should withdraw my concerns.

F. Suggested improvements: experiments, data for possible revision: I would reiterate that the authors need to be more careful in stating that the magnetised electrons likely JUST APPEAR to move faster than the unmagnetised case due to measurement limitations, rather than actually do so. Apart from that, I think at this stage of the prolonged process the material should probably be made available to the wider community for consideration on its validity. I thus recommend it be accepted for publication.

G. References: appropriate credit to previous work? This is still OK,

H. Clarity and context: lucidity of abstract/summary, appropriateness of abstract, introduction and conclusions: This is still OK, the material presented remains succinctly written.

REVIEWERS' COMMENTS (from here "R. "):

Reviewer #3 (Remarks to the Author):

Third Referee report on (revised manuscript 'Magnetic reconnection driven by electron dynamics' by Y. Kuramitsu et al.

I have read the above referenced re-revised manuscript and the rebuttal provided by the authors to my previous second report. The authors have revised one key aspect of the paper, so much of my previous report (summary, originality and significance, etc) has not changed from before. I will confine myself only to comments on the changes the authors have made since my second report.

A. Summary of the key results. As I previously reported, this paper reports a new observation which the authors claim represents magnetic reconnection occurring within a laser-produced plasma in a laboratory setting. The experimental set-up involves the production of a collimated plasma beam which propagates into a weak magnetic field. The strength of the field is such that the electrons produced in the plasma are likely to remain magnetized, while the ions are not. A key aspect of the initial paper was the interpretation of interferograms, from which the authors argued the ions appear to propagate ahead of the electrons, which deform the field. Some time after the launch of the plasma, a detached blob is detected which appears to move with the anticipated Alfvén speed of the plasma, and therefore is consistent with expectations of reconnection outflow jets and plasmoid formation.

B. Originality and significance: As I previously judged, the observations reported are original, and the overall interpretation may be correct that the morphology of the plasmas observed is related to the process of magnetic reconnection. That being the case, the observations would be of interest to the broad community of plasma physicists.

C. Data & methodology: In my previous reports I raised a concern about the interpretation, which appeared to be inconsistent with the observations presented in Figure 1, where the plasma appears to move FASTER in the presence of the magnetic field, not slower. In their original rebuttal, the authors claim that this is 'simply' due to the fact that interferometry has a certain detective range of electron density and that since the electron is collimated in the presence of the external field, the central electron density is higher than that without the external field. Thus they claim that the plasma only APPEARS to extend further in the presence of the external field, but this is not actually the case. I still feel a little unsure about drawing conclusions which rely on a non-measurement. However, it is clear from the new simulation results included in the latest revision, that it is in fact possible to identify a density contour, which, if it is the limit of measurement, would explain the observations. However, the authors still need to be very careful in their text to make it clear that in this scenario the electron plasma in the magnetic field does NOT extend further than the non-magnetic field case. This is completely evident in the new Figure 6, in which the simulations show that the magnetized electrons are significantly retarded compared to the unmagnetised case. This only appears so as an purported artefact of the measurement set-

up. The new plot in Figure 7 is also very useful in demonstrating that the magnetised plasma does not move faster than the unmagnetised case.

Authors:

We appreciate the reviewer's comment that improved our paper. We follow the comment and added the following sentences in the Discussion.

"In this scenario, the electron plasma in the presence of the magnetic field does not extend further than the case in the absence of the magnetic field. This is completely evident in Supplementary Figure 2, where the simulations show that the magnetized electrons are significantly retarded compared with the unmagnetized case. Supplementary Figure 3 also demonstrates that the magnetized plasma does not move faster than the unmagnetized case."

Reviewer:

D. Appropriate use of statistics and treatment of uncertainties: OK.

E. Conclusions: robustness, validity, reliability: As said above, the authors have demonstrated in their new simulation material that there is a defensible interpretation for the non-measurement of the electrons in the unmagnetised case. Although I find it somewhat unsatisfactory to explain the observations with the claim of an 'inability to measure', it seems to me that the authors have addressed the issue and I should withdraw my concerns.

F. Suggested improvements: experiments, data for possible revision: I would reiterate that the authors need to be more careful in stating that the magnetised electrons likely JUST APPEAR to move faster than the unmagnetised case due to measurement limitations, rather than actually do so. Apart from that, I think at this stage of the prolonged process the material should probably be made available to the wider community for consideration on its validity. I thus recommend it be accepted for publication.

G. References: appropriate credit to previous work? This is still OK,

H. Clarity and context: lucidity of abstract/summary, appropriateness of abstract, introduction and conclusions: This is still OK, the material presented remains succinctly written.